# RoboCasa365: A Large-Scale Simulation Framework for Training and Benchmarking Generalist Robots

**Soroush Nasiriany**[1,*]**, Sepehr Nasiriany**[1,*]**, Abhiram Maddukuri**[1,*]**, Yuke Zhu**[1,2]
[1]The University of Texas at Austin, [2]NVIDIA Research; * Equal contribution
https://robocasa.ai

## Abstract

Recent advances in robot learning have accelerated progress toward generalist robots that can perform everyday tasks in human environments. Yet it remains difficult to gauge how close we are to this vision. The field lacks a reproducible, large-scale benchmark for systematic evaluation. To fill this gap, we present RoboCasa365, a comprehensive simulation benchmark for household mobile manipulation. Built on the RoboCasa platform, RoboCasa365 introduces 365 everyday tasks across 2,500 diverse kitchen environments, with over 600 hours of human demonstration data and over 1600 hours of synthetically generated demonstration data—making it one of the most diverse and large-scale resources for studying generalist policies. RoboCasa365 is designed to support systematic evaluations for different problem settings, including multi-task learning, robot foundation model training, and lifelong learning. We conduct extensive experiments on this benchmark with state-of-the-art methods and analyze the impacts of task diversity, dataset scale, and environment variation on generalization. Our results provide new insights into what factors most strongly affect the performance of generalist robots and inform strategies for future progress in the field.

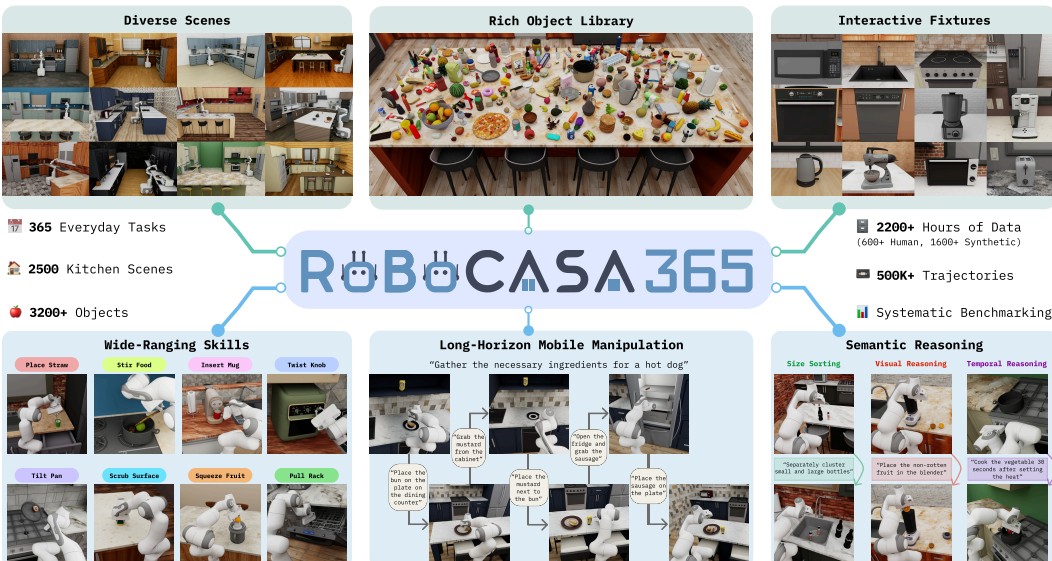

Figure 1: **Overview of RoboCasa365.** RoboCasa365 is a large-scale simulation framework for training and benchmarking generalist robots. RoboCasa365 includes 365 everyday tasks, 2500 diverse kitchen scenes, over 600 hours of human demonstration data, plus 1600 hours of synthetically generated demonstration data, and systematic benchmarks for training and evaluating generalist robot models.

# 1 INTRODUCTION

Recent advances in robot learning have brought the field closer to generalist robots capable of performing a broad range of tasks across diverse environments. A growing body of work has focused on collecting large-scale real-world robot datasets and training high-capacity robotic foundation models (Black et al., 2024; Gemini Robotics Team et al., 2025; NVIDIA et al., 2025; Physical Intelligence et al., 2025). These models have demonstrated meaningful generalization to novel objects, environments, and tasks, showing great promise towards developing broadly capable policies.

Despite these advances, two major challenges remain. First, training generalist robots requires vast amounts of robot experience data. Although recent datasets have grown substantially in size, they remain limited in diversity and task coverage, which constrains the ability to train robust, generalist policies. Second, real-world evaluation and benchmarking are resource-intensive and time-consuming. They are often affected by experimental noise, making it difficult to perform reproducible, systematic comparisons across methods.

Simulation provides a practical avenue for addressing these challenges. With simulation, we can create large-scale interaction datasets, covering an effectively infinite variety of tasks and environments (Mandlekar et al., 2023; Jiang et al., 2025). Simulation also enables rapid experimentation, controlled evaluation, and reproducible benchmarking that would be infeasible in real-world robotics (Saxena et al., 2025). Together, these capabilities make it possible to generate data, train policies, and systematically evaluate generalist robots at scale. However, existing simulation frameworks fall short of this potential. Most current tools support only limited tasks and environments, often focusing on simple object manipulation or single-room scenarios (Zhu et al., 2020; James et al., 2020; Wang et al., 2023). The datasets they generate are small relative to the diversity and complexity of real-world robotics challenges, and benchmarking is typically confined to these narrow conditions (Liu et al., 2023; Mandlekar et al., 2021). Consequently, it remains difficult to study how task diversity, environment variation, and dataset scale affect policy generalization.

To address these gaps, we introduce RoboCasa365, a comprehensive simulation benchmark for everyday household robotics. RoboCasa365 is built on top of the RoboCasa simulation framework by Nasiriany et al. (2024), and is structured around four core components:

**Comprehensive tasks**: RoboCasa365 defines 365 tasks spanning 60 distinct kitchen activities, including manipulation, semantic reasoning, long-horizon planning, and memory-dependent tasks. This task diversity allows evaluation across multiple dimensions of generalist robot capability.

**Diverse environments**: The benchmark includes 2,500 unique kitchen scenes modeled from real kitchens across the United States. These scenes capture a wide spectrum of layouts, object configurations, and visual variations, providing realistic contexts for a variety of everyday tasks.

**Large-scale data**: The benchmark provides over 2,000 hours of robot interaction data. This includes 612 hours of human demonstration data and an additional 1615 hours of synthetic demonstration data using the MimicGen data generation tool (Mandlekar et al., 2023) to significantly expand the quantity of data.

**Systematic benchmarking**: RoboCasa365 supports rigorous evaluation across three learning settings: massively multi-task training, foundation model training, and lifelong learning. The benchmark is designed to facilitate reproducible, large-scale experiments and in-depth analysis of which data and environment factors most strongly influence generalization.

By integrating these elements, RoboCasa365 provides a large, diverse, and systematically structured resource for studying generalist robots in simulation. It enables researchers to explore algorithms, run reproducible evaluations, and analyze the impact of task and environment diversity on policy generalization. Using RoboCasa365, we conduct extensive experiments to compare state-of-the-art methods, evaluate learning strategies, and investigate the factors that most strongly drive performance in generalist robot learning.

# 2 RELATED WORK

**Robot Simulation Frameworks.** There is a long line of prior work on building robot simulation frameworks (Zhu et al., 2020; Gu et al., 2023; Mittal et al., 2023; Tao et al., 2025; Szot et al., 2021;

Kolve et al., 2017; Li et al., 2023; 2024; Liu et al., 2023; Deitke et al., 2022). Some are focused on tabletop settings (Zhu et al., 2020; Liu et al., 2023; Li et al., 2024; James et al., 2020), while we focus on simulating entire room-scale scenes, similar to some other prior works (Li et al., 2023; Nasiriany et al., 2024; Szot et al., 2021; Kolve et al., 2017). Our work is unique in that it features hundreds of tasks across thousands of unique scenes, large-scale, high-quality demonstration datasets, and a suite of benchmarks for training and evaluating generalist robot models. To our best knowledge, our work is the first simulation framework to satisfy all of these criteria.

**Datasets and Benchmarks for Generalist Robots.** There have been numerous efforts towards collecting large robot datasets in the real world (Brohan et al., 2022; Walke et al., 2023; Khazatsky et al., 2024; Open X-Embodiment Collaboration et al., 2023). Evaluating and benchmarking policies trained on these datasets in the real world is challenging due to the resources needed to run large-scale systematic evaluations, despite several recent approaches towards this goal (Atreya et al., 2025; Zhou et al., 2023; Yenamandra et al., 2023; Zhou et al., 2025; Krotkov et al., 2016; Correll et al., 2018). Simulation enables running large-scale benchmarks. However, most simulation benchmarks are confined to a very narrow distribution of tasks and environments (Mandlekar et al., 2021; Zhu et al., 2020; Liu et al., 2023; TRI LBM Team et al., 2025). Li et al. (2023) bring forth some of the largest diversity of environments and tasks to date, but lack accompanying large-scale datasets for all of these tasks. Nasiriany et al. (2024) include 100k demonstrations spanning 30 tasks and 100 scenes. In contrast, our datasets comprise over 500k demonstrations across over 300 tasks and 2500 unique scenes. While prior work focuses on benchmarking specific methods such as multi-task training (TRI LBM Team et al., 2025; Nasiriany et al., 2024) and lifelong learning (Liu et al., 2023), we provide a comprehensive suite of benchmarks to systematically study multi-task training, foundation model training, and lifelong learning.

**Training Generalist Robots.** There is a long body of work on learning generalist robot policies from large, diverse robot datasets (Octo Model Team et al., 2024; Open X-Embodiment Collaboration et al., 2023; NVIDIA et al., 2025; Brohan et al., 2023; Kim et al., 2024; Shukor et al., 2025; Wen et al., 2025). In our work, we aim to be agnostic to the choice of model, and instead create benchmarks to systematically assess the capabilities of these models across distinct settings, including multi-task training, pretraining, and fine-tuning on target data, and lifelong learning.

## 3 RoboCasa365: Large-Scale Simulation of 365 Everyday Tasks

We present RoboCasa365, a large-scale simulation framework for training and benchmarking generalist robots. We use the existing RoboCasa simulation framework (Nasiriany et al., 2024) as the starting ground for RoboCasa365 and make significant efforts to scale up the assets, environments, tasks, and datasets. We also establish a rigorous benchmark to study state-of-the-art policy learning methods, which we outline in Section 4. In the following sections, we outline the components of this simulation framework: assets, scenes, tasks, and datasets.

### 3.1 Expanding the Scope of Assets

RoboCasa features a diverse array of objects, interactable fixtures, and appliances, with a focus on common tasks in kitchen environments. We use the existing library of 2,509 objects from Nasiriany et al. (2024), spanning 153 object categories. In addition to these, we source an additional collection of high-quality 3D assets spanning 57 object categories. These are high-quality 3D assets sourced from artists and edited to preserve strict quality standards. We use these new objects to support new tasks and to populate various areas of kitchen scenes generally. We provide a complete inventory in Appendix C.1.

In addition to the 3D assets, we significantly expand the repertoire of interactable fixtures and appliances in the kitchen environment. RoboCasa (Nasiriany et al., 2024) includes a total of 20 interactable fixtures and appliances across 4 categories: sinks, coffee machines, stoves, and microwaves. We significantly expand the scope of these assets to 456 instances spanning 12 categories. We include new categories of appliances, such as toasters, toaster ovens, stand mixers, blenders, and electric kettles. All of these appliances are articulated, including fridges, ovens, and dishwashers, which were not previously articulated under RoboCasa. We model these assets using the same format as RoboCasa, as MJCF objects with annotations of the regions. For each category, we include between

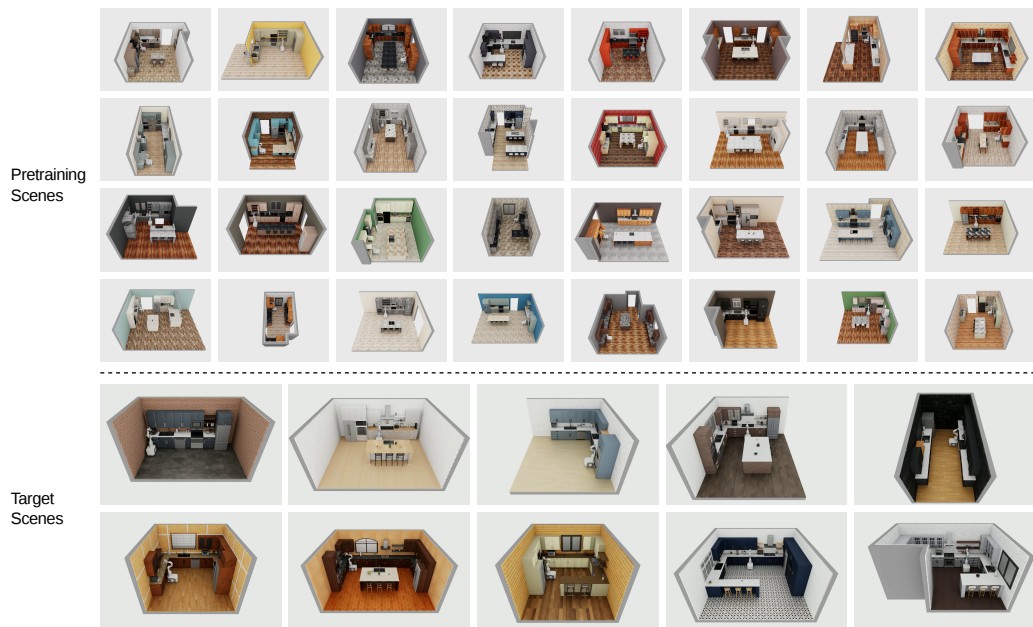

Figure 2: **Kitchen Scenes.** Our simulation framework features 2,500 distinct kitchen scenes for pretraining (top, representative samples shown), and 10 distinct target kitchen scenes (bottom, all scenes shown).

20 and 50 instances in order to ensure that there is sufficient diversity to support generalization to novel instances. We provide a complete inventory of our fixtures and appliances in Appendix C.2.

## 3.2 DIVERSE KITCHEN SCENES

Achieving generalization in robot learning requires exposure to a wide range of training environments; we address this need by providing thousands of diverse kitchen scenes spanning a broad spectrum of household settings. We categorize these scenes into *pretraining* and *target* splits. Our goal is to use the pretraining kitchen scenes for large-scale data collection and synthetic data generation pipelines; we use the target kitchen scenes for targeted data collection and for running most of our experiment evaluations. Using the terminology from Nasiriany et al. (2024), we define each kitchen scene as a combination of *layout* and *style*, where the layout defines the floor plan, and the style defines the specific selection of fixtures, appliances, and textures used in the kitchen. We can configure each kitchen scene to use any combination of layout and scene.

For our target kitchens, we use the 10 layouts and 10 styles defined by Nasiriany et al. (2024) in RoboCasa, where each layout is matched with a specific style, for a total of 10 kitchen scenes. For our pretraining kitchen scenes, we create 50 distinct new layouts. In order to capture the distribution of diverse scenes, we source our kitchens from 50 real-world homes with active listings on Zillow.com, a real estate marketplace. These homes span diverse geographic locations across the United States. We build digital cousin (Dai et al., 2024) replicas for each of these environments, making sure to match the floor plan as closely as possible. In addition to these layouts, we create 50 distinct styles. We ensure that the pretraining and target styles do not overlap in the selection of the fixtures, appliances, or environment textures used. Together, we have a total combination of 50 layouts × 50 styles, for a total of 2,500 pretraining kitchen scenes. We provide an overview of the pretraining and target kitchen scenes in Figure 2.

## 3.3 SUITE OF 365 EVERYDAY TASKS

We aim to provide a diverse set of tasks to support sharing knowledge across tasks and generalizing to new tasks. Nasiriany et al. (2024) define two broad categories of tasks: *atomic tasks*, which represent the execution of a single skill, and *composite tasks*, which involve executing a sequence of skills. Nasiriany et al. (2024) define eight foundational skills: (1) pick-and-place, (2) opening and

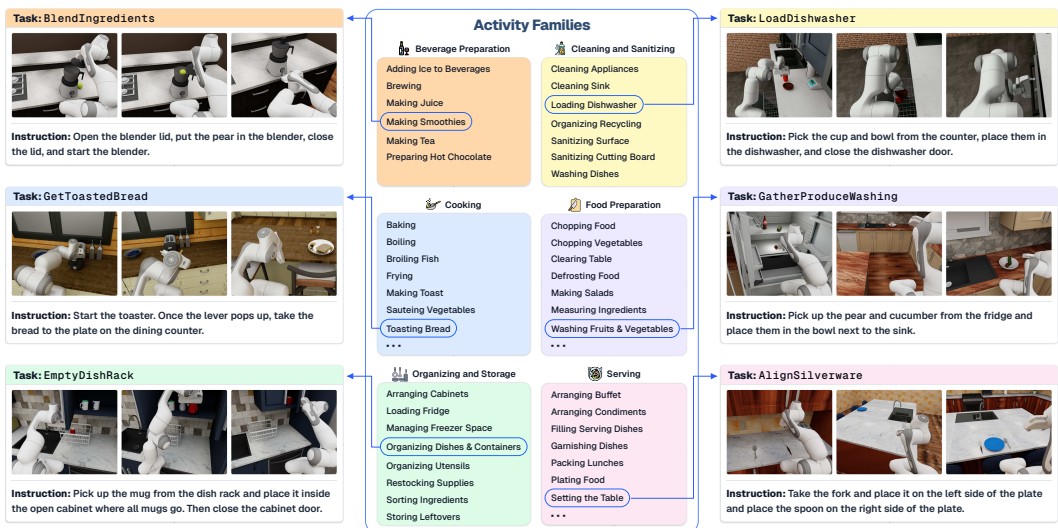

Figure 3: **Composite Tasks.** RoboCasa365 features 300 composite tasks that involve a sequence of skills. We use large language models to generate a set of high-level activities, and for each activity, a set of task blueprints. There are 6 activity families (high-level categories) spanning 60 activities, which organize composite tasks based on shared functional and semantic structure. Representative tasks are shown for selected activities.

closing doors, (3) opening and closing drawers, (4) turning levers, (5) turning knobs, (6) pressing buttons, (7) insertion, and (8) navigation. We adopt these skills as the basis for our atomic tasks. In addition to the 25 atomic tasks in RoboCasa, we create an additional set of 40 new atomic tasks to support various new appliances and new behaviors afforded by our simulator. We provide the entire list of 65 atomic tasks in Appendix E.1.

For our composite tasks, we follow the framework established by Nasiriany et al. (2024), where we use large language models to solicit task blueprints. The process follows two stages. First, we prompt LLMs to give a list of *activities* representing high-level groups of tasks in kitchen environments. We retrieve a list of the top 60 activities, such as boiling water, toasting bread, brewing coffee, washing dishes, and storing leftovers, to name a few. For each activity, we then prompt the LLM to provide task blueprints, which consist of the name of the task, a high-level description of the task, the objects and fixtures involved, and the sequences of skills needed to solve the task. We then proceed to write code for the tasks based on these blueprints. We use 83 of the existing composite tasks from RoboCasa and generate an additional set of 217 new composite tasks, for a total of 300 composite tasks. We outline the full list of activities and representative composite tasks in Figure 3. In total, our benchmark includes 365 everyday tasks: 65 atomic tasks and 300 composite tasks. Out of these, 220 require mobile manipulation, while 145 can be performed without mobility.

## 3.4 DATASETS

We provide a large collection of robot datasets covering all of our tasks. Broadly, our datasets are divided into two categories: *pretraining datasets* for data from the pretraining scenes, and *target datasets* from the target scenes.

### 3.4.1 PRETRAINING DATASETS

Out of the 365 total tasks outlined in Section 3.3, our pretraining data covers 300 tasks, with 65 atomic tasks and 235 composite tasks. For each of these 300 tasks, we collect 100 human demonstrations per task via robot teleoperation. This results in 30k human demonstrations total for pretraining. For our data collection, we use the Franka Panda Emika robot, equipped with an Omron mobile base (Haviland et al., 2022), and in principle, our simulation framework can support data collection with other mobile manipulators and humanoid platforms.

We also use the MimicGen generation system (Mandlekar et al., 2023) to generate large-scale synthetic data across 60 atomic tasks. For each task, we use the 100 human demonstrations previously collected as seed demonstrations, and generate 10k demonstrations, effectively scaling data 100×.

### 3.4.2 TARGET DATASETS

For our target data, we choose 50 representative ones out of the 365 tasks, grouped into three splits:

- `Atomic` (18 tasks): We include 18 representative tasks out 65 total atomic tasks in the benchmark.
- `Composite-Seen` (16 tasks): We choose 16 representative composite tasks spanning 16 activities. These include a mix of short and long-horizon tasks, with some involving 2 subtasks and the longest task involving 15 subtasks.
- `Composite-Unseen` (16 tasks): To test the effect of our pretraining data, we also choose 16 composite tasks that are unseen in the pretraining data. These tasks are of similar difficulty to the composite seen tasks, but focus on another distinct set of 16 activities.

We list the entire set of 50 target tasks in Appendix E.2. For each of these tasks, we collect 500 human demonstrations via robot teleportation, for a total of 25k demonstrations.

### 3.4.3 DATASET STATISTICS

We provide a high-level overview of our datasets in Appendix F. Our pretraining synthetic demonstration dataset spans the highest amount of data, with 1615 total hours, followed by human pretraining data (404 hours), and then human target data (208 hours). In Figure 4a we report the distribution over the number of subtasks required for each of our 365 tasks. Most tasks require one or two subtasks, but there are a few tasks that require 15 or more subtasks to complete. In Figure 4b, we report the distribution of episode lengths across all pretraining and target human data (55k episodes). The majority of episodes range from 10 to 60 seconds, with a long tail end for longer horizon episodes, some going beyond 3 minutes.

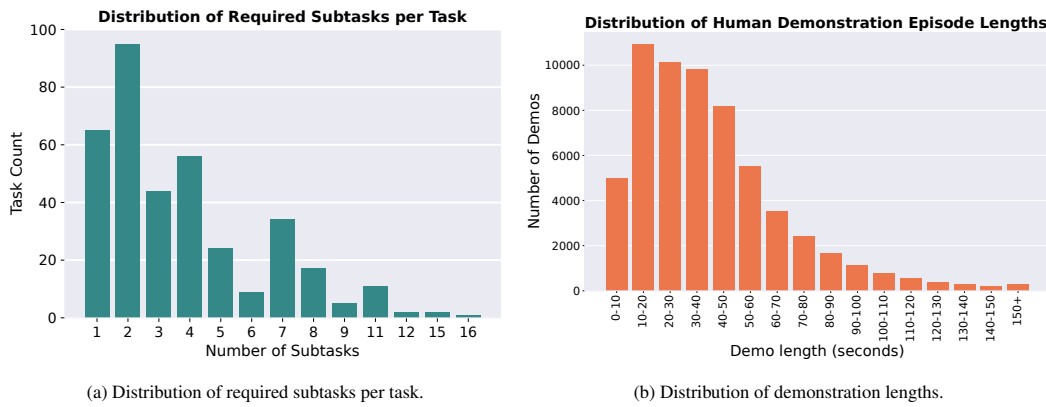

(a) Distribution of required subtasks per task.     (b) Distribution of demonstration lengths.

Figure 4: Distribution of task lengths (by number of subtasks) and dataset episode lengths (by number of seconds). We observe a long tail of tasks and data representing long-horizon behaviors.

## 4 EXPERIMENTS

In our experiments, we conduct a systematic study to understand the key factors that influence training generalist robot policies. To this end, we design a comprehensive suite of benchmarks aimed at answering the following questions:

1. How well do generalist robot models perform when trained on large multi-task datasets?
2. What role does pretraining data play, and to what extent can it improve learning of downstream tasks?

3. How effectively can we learn new tasks in lifelong learning settings?
4. How does the scope and composition of pretraining data impact learning downstream tasks?

## 4.1 Multi-task training

We begin by investigating how state-of-the-art methods perform when trained on massively multi-task datasets. This evaluation is a critical step toward developing generalist robots that can not only master a wide range of behaviors but also adapt to entirely novel tasks beyond their training data.

We train language-conditioned vision-based policies on the mixture of 300 pretraining human datasets outlined in Section 3.4.1. Each task has 100 human demonstrations, for a total of 30k demonstrations. Our experiments feature four state-of-the-art methods: **Diffusion policy** (Chi et al., 2023), $\pi_0$ (Black et al., 2024), $\pi_{0.5}$ (Physical Intelligence et al., 2025), and **GR00T N1.5** (NVIDIA et al., 2025).

We train a multi-task language-conditioned policy for each method. We use the pretrained checkpoints released publicly for $\pi_0$, $\pi_{0.5}$, and GR00T N1.5 as the base model for training our models. We provide details on the training protocols for each method in Appendix G.

We evaluate on the 50 tasks outlined in Section 3.4.2: `Atomic`, `Composite-Seen`, and `Composite-Unseen`. Note that the `Composite-Unseen` tasks represent unseen tasks in the pretraining data; our evaluation for these tasks is zero-shot, aimed at understanding generalization to novel tasks. We evaluate in the *pretraining* kitchen scenes for each task and report average task completion success rates across methods. See Appendix G for details on the evaluation protocol.

| Task Split | Diffusion Policy | $\pi_0$ | $\pi_{0.5}$ | GR00T N1.5 |
|---|---|---|---|---|
| `Atomic` | 15.7 | 36.3 | 39.6 | 43.0 |
| `Composite-Seen` | 0.2 | 5.2 | 7.1 | 9.6 |
| `Composite-Unseen` | 1.25 | 0.7 | 1.2 | 4.4 |
| Average | 6.1 | 15.0 | 16.9 | 20.0 |

Table 1: **Multi-task Training Results.** We compare state-of-the-art policy learning approaches on our human pretraining data across 300 tasks, and report task success rates (%) across seen and unseen tasks. We see that learning composite tasks is more challenging, and that performance suffers when evaluating on unseen tasks.

We report results in Table 1. Overall, we see that across all methods, learning `Atomic` tasks is the easiest, followed by learning `Composite-Seen` tasks and `Composite-Unseen` tasks. This is reasonable, as the `Atomic` tasks are shorter-horizon tasks that present fewer learning challenges for imitation learning (Ross et al., 2011), and the lower performance on `Composite-Unseen` tasks is due to the fact that the model has never been trained on these tasks. Overall, GR00T N1.5 performs the best among all methods, followed by $\pi_{0.5}$, $\pi_0$, and finally Diffusion Policy. They show non-zero success rates on `Composite-Unseen` tasks, a sign of stronger generalization abilities. Diffusion Policy performs the worst, highlighting how high-capacity vision-language-action models can better fit large, diverse multi-task robot datasets. While our multi-task learning experiments show that GR00T N1.5 outperforms other baselines, we do not claim that it is conclusively the superior method. Performance can be influenced by many factors, including the amount of compute used (*e.g.*, batch size), data composition, and whether the visual or language backbones are fine-tuned. Overall, we see a significant opportunity for future methods to improve upon these results.

## 4.2 Foundation model training

In our next experiment, we are interested in studying foundation model training, *i.e.*, training with our pretraining datasets, followed by fine-tuning on our target datasets. This learning paradigm has been established by numerous prior works in robotics (Black et al., 2024; NVIDIA et al., 2025), with evidence that pretraining can aid learning downstream tasks in a more robust and data-efficient manner. In our experiments, our pretraining data includes human datasets across 300 tasks (411 hours), and synthetic data across 60 atomic tasks (1,615 hours), while our target data includes human datasets across 50 tasks (208 hours). Out of the 50 target tasks, 34 are also represented in the human pretraining data (`Atomic` and `Composite-Seen` tasks), and the target data includes an additional 16 composite tasks that are not seen in the pretraining data (`Composite-Unseen`). We

| Task Type | Pretraining Only | Target Only | | | Pretraining + Target Post-Training | | |
|---|---|---|---|---|---|---|---|
| | | 10% | 30% | 100% | 10% | 30% | 100% |
| Atomic | 41.9 | 38.7 | 50.6 | 60.6 | 56.9 | 59.1 | **68.5** |
| Composite-Seen | 0.0 | 11.0 | 22.7 | 35.0 | 25.4 | 34.6 | **40.6** |
| Composite-Unseen | 0.2 | 11.2 | 27.5 | 33.3 | 22.7 | 30.8 | **42.1** |
| Average | 15.1 | 21.0 | 34.3 | 43.7 | 35.9 | 42.2 | **51.1** |

Table 2: **Foundation Model Training Results.** Comparing the impact of training on pretraining and target datasets on learning downstream tasks. The performances are measured by average task success rates (%).

first train on all of our pretraining datasets (see Section 3.4.1), followed by fine-tuning independently on three separate target split datasets (`Atomic`, `Composite-Seen`, `Composite-Unseen`; see Section 3.4.2). We compare learning on different amounts of target data, with 50, 150, and 500 demos per task, representing 10%, 30%, and 100% of the total target data.

**Unless otherwise noted, we use GR00T N1.5 as the model for these experiments and all subsequent experiments.** We open source all models for the community to benchmark all methods. We compare pretraining only, target task learning only, and pretraining followed by post-training on target data. After training, we evaluate the model across the 50 target tasks in the target kitchens. See Appendix G for a detailed discussion of the training and evaluation protocols. We report experiment results in Table 2.

We see that with pretraining alone, the model performs over 40% on the atomic tasks but performs very poorly on the composite tasks. For target learning only, we see more capable policies. However, they require a high amount of data to be performant. Using pretraining yields significant improvements in model performance. These gains are especially pronounced for the `Composite-Unseen` tasks (see Table 2). We visualize the improvement in performance in Figure 5, visualizing the average task success rates from Table 2. We observe a roughly $3\times$ improvement in data efficiency, *i.e.*, pretraining helps achieve roughly the same performance as target learning only with $3\times$ higher number of target task demonstrations. In Appendix H.2, we present a rigorous robustness evaluation and analyze the effects of different factors on performance.

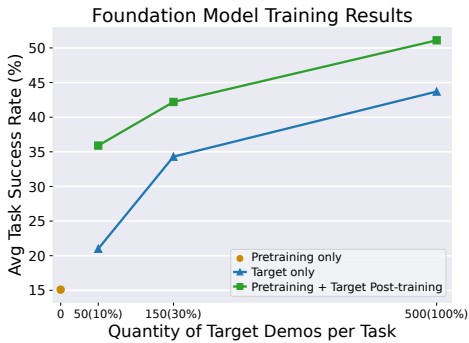

Figure 5: **Foundation Model Training Results.** Pre-training enables more effective learning of downstream tasks with significant gains in data efficiency.

## 4.3 LIFELONG LEARNING

In contrast to the conventional two-stage paradigm of pretraining followed by post-training on target data, real-world robots must often acquire new skills continuously. This setting, known as lifelong learning, involves learning tasks over a sequence of phases. The central challenge is leveraging prior knowledge to learn new tasks while retaining previously acquired skills. We design a lifelong learning benchmark to assess these capabilities. In our experiments, we learn a series of tasks over four phases. Each phase involves learning progressively longer horizon tasks. Phase 1 involves learning 65 atomic tasks, Phase 2 involves learning 20 new composite tasks with 2 or 3 stages, Phase 3 involves learning 20 new composite tasks with 4 or 5 stages, and Phase 4 involves learning 20 new composite tasks with 6 or more stages. We define "stage" as the invocation of one of the robot skills defined by Nasiriany et al. (2024), such as pick-and-place, turning knobs, and navigation. We use pretraining datasets for these phases; Phase 1 includes all human and MimicGen datasets for atomic tasks, while Phases 2, 3, and 4 feature human datasets.

For each phase $N$, we take the model previously trained from phase $N-1$, and fine-tune it for data pertaining to the tasks in phase $N$. After training completes for phase $N$, we run evaluations for tasks from phase 1 through phase $N$ in the pretraining kitchens and report results. We report results in Table 3. We make two distinct observations. First, we see that the success rates steadily drop as we learn progressively longer-horizon tasks in each new phase (see the diagonal entries in the table).

| Phase | Atomic Tasks | 2-3 Stage Tasks | 4–5 Stage Tasks | 6+ Stage Tasks |
|-------|--------------|-----------------|-----------------|----------------|
| Phase 1 | 41.5 | - | - | - |
| Phase 2 | 13.9 | 24.5 | - | - |
| Phase 3 | 13.9 | 4.8 | 11.3 | - |
| Phase 4 | 10.6 | 1.7 | 2.7 | 4.3 |

Table 3: **Lifelong Learning Results.** We train across four phases with progressively longer horizon tasks. After each phase, we report task success rates (%) across all tasks seen in the current and previous phases.

This is intuitively the case, as learning longer-horizon tasks can demand higher data requirements. Second, we see that the performance on previously learned tasks steadily drops with each new phase. This highlights the *catastrophic forgetting problem*, *i.e.*, performance degrades on prior tasks if the agent does not continue to train on them in subsequent phases. Overall, this experiment highlights the current challenges with lifelong learning and is a useful testbed for improving upon these results.

### 4.4 PRETRAINING DATA COMPOSITION STUDY

In Section 4.2, we showed that pretraining brings forth significant improvements in data efficiency for learning downstream target tasks. In this section, we run experiments to further understand how the composition of pretraining data affects downstream performance. In our foundation model training experiments, we used all of the available pretraining data, comprising human data from 300 tasks and MimicGen data across 60 tasks (Human300 + MG60). We compare to a variant that does not include MimicGen data and only includes the human data (Human300). To better understand the role of task diversity in the pretraining data, we compare two variants that include human data from 50 tasks (Human50). These 50 tasks include the `Atomic` and `Composite-Seen` tasks, as well as an additional randomly selected set of tasks. Finally, we compare with the case with no pretraining data. Our pretraining and target protocol are identical to the process in Section 4.2. We specifically run two separate sets of experiments, one for the low-data regime with 10% of the target data, and one for the high-data regime with 100% of the target data.

| Target Data | Pretraining Data | | | |
|-------------|------------------|----------|-----------|------------------|
| | No Pretraining | Human50 | Human300 | Human300 + MG60 |
| Atomic (10%) | 38.7 | 52.0 | **57.0** | 56.9 |
| Composite-Seen (10%) | 11.0 | 26.2 | **28.7** | 25.4 |
| Composite-Unseen (10%) | 11.2 | 23.8 | **32.3** | 22.7 |
| Average (10%) | 21.0 | 34.7 | **40.0** | 35.9 |
| Atomic (100%) | 60.6 | 68.1 | **70.0** | 68.5 |
| Composite-Seen (100%) | 35.0 | 41.0 | **41.2** | 40.6 |
| Composite-Unseen (100%) | 33.3 | 38.5 | **44.0** | 42.1 |
| Average (100%) | 43.7 | 50.0 | **52.5** | 51.1 |

Table 4: **Pretraining Task Diversity Results.** We report task success rates (%) and compare the downstream effects of training on different mixtures of pretraining data.

We report evaluations in target kitchens in Table 4. Compared to training on all pretraining data (Human300 + MG60), we find that training on just the human data (Human300) yields better downstream learning results. Although MimicGen enables the large-scale generation of synthetic trajectories, we find that the resulting demonstrations vary in quality. Developing methods that can more effectively leverage such large, mixed-quality datasets is an important direction for future work. Comparing the Human50 and Human300 settings, we see that increasing the number of pretraining tasks can enable a significant improvement in downstream target tasks, especially for the low-data regime target data setting. Notably, the biggest gains are seen for the `Composite-Unseen` tasks, suggesting that increasing the scope of task diversity is especially beneficial for learning novel tasks.

In addition to task diversity, we study the effects of scene diversity in pretraining on downstream performance. We report these results in Appendix H.1.

### 4.5 REAL-WORLD EXPERIMENTS

We conduct an additional set of experiments to examine the utility of our benchmark for downstream real-world applications. Our real-world setup uses the DROID Panda arm (Khazatsky et al., 2024) with three cameras.

We examine four tasks in a real kitchen:

- `CloseElectricKettleLid`: close the electric kettle lid
- `PickPlaceToasterOvenToCounter`: place the item from the toaster oven to the counter
- `PickPlaceCounterToCabinet`: place the object from the counter to the cabinet
- `PlaceOnDishRack`: a longer horizon task, involving placing two items from the sink onto the dish rack.

We collect 30 demonstrations for each of the first three tasks, and 50 demonstrations of the last task, for a total of 140 real-world demonstrations. We compare the following settings:

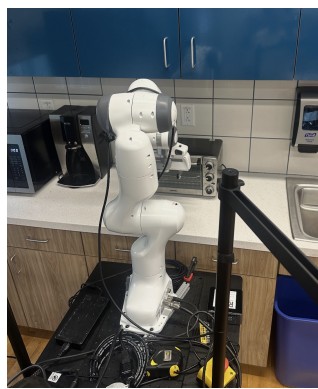

- Real Only: we train the GR00T N1.5 model on the real-world demonstrations (140 demonstrations);
- Sim-and-Real (Ours): we first mid-train the GR00T N1.5 model on our simulation tasks (we use data from the 150 highest performing tasks in simulation), and then co-fine-tune the model on the real-world demonstrations and corresponding data for the four real-world tasks in simulation.

Following best practices for sim-and-real alignment from Maddukuri et al. (2025), we re-render our simulation datasets to match the camera views of the real setting, facilitating improved transfer. After training, we evaluate each model in the real world, where we conduct 20 trials per task. We report task success rates in Table 5.

Figure 6: **Real-Robot Platform.** Our real-world setup features a Panda robot arm in a real kitchen.

| | CloseElectric KettleLid | PickPlaceToasterOven ToCounter | PickPlaceCounter ToCabinet | PlaceOn DishRack | Avg |
|---|---|---|---|---|---|
| Real Only | **70** | 70 | 52 | 55 | 61.8 |
| Sim-and-Real (Ours) | **70** | **100** | **84** | **65** | **79.8** |

Table 5: **Real-world evaluations.** Across four real-world tasks, we compare training on real-world data only versus training on a mixture of our simulation and real-world data. By additionally using simulation data, it outperforms training on real-world data only by an average task success rate of 18.1%.

Overall, the Real Only model achieves a 61.8% average success rate, while Sim-and-Real training reaches 79.8%, a substantial improvement. This highlights the value of our simulation benchmark for both algorithm evaluation and real-world policy learning.

## 5 CONCLUSION

We presented RoboCasa365, a large-scale simulation framework for training and benchmarking generalist robot models. RoboCasa365 provides 2,500 realistic kitchen environments, 365 everyday tasks spanning over 50 activity categories, and over 2,000 hours of robot interaction data, making it one of the most diverse simulation resources to date.

Using this benchmark, we conducted a systematic study along three axes: multi-task learning at scale, foundation model learning, and lifelong learning. Our experiments show that (i) generalist policies trained on large multi-task datasets can acquire broad competence but still face challenges with long-horizon tasks, (ii) pretraining data significantly improves downstream learning, with both scale and task diversity playing key roles, and (iii) lifelong learning remains an open challenge, with substantial trade-offs between acquiring new tasks and retaining prior knowledge.

RoboCasa365 opens several avenues for future work. First, the benchmark is currently limited to kitchen environments, raising the question of how well findings transfer to other household settings or broader domains. Second, while the dataset is large, it does not capture the full sensory and physical complexity of the real world, and bridging the gap between simulation and real-world deployment remains a significant challenge. Addressing these limitations will be an important direction for future research.

## ACKNOWLEDGMENTS

We thank Qi Wang for his valuable assistance in coordinating project resources, particularly in data collection and asset preparation. We thank Steve Xie and the LightWheel team for their close collaboration in providing simulation assets and support with data collection. We also thank Ajay Mandlekar, Zi-ang Cao, and Kevin Lin for their assistance with running benchmarking experiments. Part of this work was done during Soroush Nasiriany's internship at NVIDIA Research. This work was partially supported by the National Science Foundation (FRR-2145283, EFRI-2318065), the Office of Naval Research (N00014- 24-1-2550), the DARPA TIAMAT program (HR0011-24-9-0428), the Army Research Lab (W911NF-25-1-0065), and the KIST-UT collaboration (UTAUS-FA00004578). It was also supported by the Institute of Information & Communications Technology Planning & Evaluation (IITP) grant funded by the Korean Government (MSIT) (No. RS2024-00457882, National AI Research Lab Project).

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

## A  USE OF LARGE LANGUAGE MODELS

We use the aid of large language models to create activity labels and task blueprints, using the process outlined by Nasiriany et al. (2024). We also use large language models for soliciting writing feedback for parts of this manuscript.

## B  SIMULATION INFRASTRUCTURE

### B.1  PHYSICS AND RENDERING ENGINE

RoboCasa365 is built on top of RoboSuite (Zhu et al., 2020), which uses the MuJoCo physics engine (Todorov et al., 2012). While MuJoCo's core physics computations are CPU-based, we leverage GPU-based rendering. RoboCasa365 simulates at 20 Hz, with the simulation running approximately in real time, slightly faster or slower depending on scene complexity and hardware specifications. Multiple asynchronous environments can be run in parallel, allowing overall throughput to scale with the number of available CPU cores and GPUs.

### B.2  ACTION SPACE

We adopt the underlying controller from RoboCasa (Nasiriany et al., 2024). Specifically, we use an Operational Space Controller (Khatib, 1995) running at 20 Hz that commands the arm through seven action dimensions: three for translation, three for rotation, and one for gripper opening and closing. In addition, we include five action dimensions for mobile base translation and rotation, torso height control, and an action mode that gates mobile base control.

## C  SIMULATION ASSETS

### C.1  3D OBJECTS

We have added new objects across 57 categories: aluminum foil, basket, blender jug, cheese grater, chicken drumstick, cinnamon, colander, cookie dough ball, cream cheese stick, digital scale, dish brush, flour bag, glass cup, honey bottle, hotdog bun, ice cube, ice cube tray, jar, juice, kebab skewer, lemon wedge, lettuce, marshmallow, mayonnaise, measuring cup, mustard, non electric kettle, oil and vinegar bottle, oven tray, pancake, paprika, peeler, pickle slice, pitcher, pizza, pizza cutter, placemat, pot, reamer, salt and pepper shaker, sandwich bread, saucepan, saucepan lid, shrimp, soap dispenser, spray, strainer, straw, sugar cube, syrup bottle, tomato slice, tongs, tupperware, turkey slice, turmeric, whisk, and wooden spoon.

### C.2  INTERACTIVE FIXTURES AND APPLIANCES

We report an inventory of all fixtures and appliances in Table 6.

| Category | Unique models |
|---|---|
| Blender | 22 |
| Coffee machine | 48 |
| Dishwasher | 25 |
| Electric kettle | 25 |
| Fridge | 50 |
| Microwave | 50 |
| Oven | 21 |
| Sink | 49 |
| Stand mixer | 25 |
| Stove | 50 |
| Toaster | 44 |
| Toaster oven | 47 |
| **Total** | **456** |

Table 6: Inventory of fixtures and appliances

## D  SCENES

We build 50 kitchen layouts modeled after 50 homes on sale on Zillow.com. These homes span locations in the Bay Area (California), Austin (Texas), Denver (Colorado), Boston (Massachusetts), and Atlanta (Georgia).

## E  TASKS

### E.1  ATOMIC TASKS

We have 65 atomic tasks: `AdjustToasterOvenTemperature`, `AdjustWaterTemperature`, `CheesyBread`, `CloseBlenderLid`, `CloseCabinet`, `CloseDishwasher`, `CloseDrawer`, `CloseElectricKettleLid`, `CloseFridge`, `CloseFridgeDrawer`, `CloseMicrowave`, `CloseOven`, `CloseStandMixerHead`, `CloseToasterOvenDoor`, `CoffeeServeMug`, `CoffeeSetupMug`, `LowerHeat`, `MakeIcedCoffee`, `NavigateKitchen`, `OpenBlenderLid`, `OpenCabinet`, `OpenDishwasher`, `OpenDrawer`, `OpenElectricKettleLid`, `OpenFridge`, `OpenFridgeDrawer`, `OpenMicrowave`, `OpenOven`, `OpenStandMixerHead`, `OpenToasterOvenDoor`, `PackDessert`, `PickPlaceCabinetToCounter`, `PickPlaceCounterToBlender`, `PickPlaceCounterToCabinet`, `PickPlaceCounterToDrawer`, `PickPlaceCounterToMicrowave`, `PickPlaceCounterToOven`, `PickPlaceCounterToSink`, `PickPlaceCounterToStandMixer`, `PickPlaceCounterToStove`, `PickPlaceCounterToToasterOven`, `PickPlaceDrawerToCounter`, `PickPlaceFridgeDrawerToShelf`, `PickPlaceFridgeShelfToDrawer`, `PickPlaceMicrowaveToCounter`, `PickPlaceSinkToCounter`, `PickPlaceStoveToCounter`, `PickPlaceToasterOvenToCounter`, `PickPlaceToasterToCounter`, `PreheatOven`, `SlideDishwasherRack`, `SlideOvenRack`, `SlideToasterOvenRack`, `StartCoffeeMachine`, `TurnOffMicrowave`, `TurnOffSinkFaucet`, `TurnOffStove`, `TurnOnBlender`, `TurnOnElectricKettle`, `TurnOnMicrowave`, `TurnOnSinkFaucet`, `TurnOnStove`, `TurnOnToaster`, `TurnOnToasterOven`, and `TurnSinkSpout`.

### E.2  TARGET TASKS

We provide an overview for the target tasks across Tables 7, 8, and 9.

| Activity | Task | # Sub-tasks | MoMa req. | Description |
|---|---|---|---|---|
| Atomic | `CloseBlenderLid` | 1 | No | Close the lid blender by securely placing the lid on top. |
| Atomic | `CloseFridge` | 1 | No | Close the fridge door(s). |
| Atomic | `CloseToasterOvenDoor` | 1 | No | Close the toaster oven door. |
| Atomic | `CoffeeSetupMug` | 1 | No | Pick the mug from the counter and place it under the coffee machine dispenser. |
| Atomic | `NavigateKitchen` | 1 | Yes | Navigate to the [*kitchen location*]. |
| Atomic | `OpenCabinet` | 1 | No | Open the cabinet door(s). |
| Atomic | `OpenDrawer` | 1 | No | Open the [*left/right*] drawer. |
| Atomic | `OpenStandMixerHead` | 1 | No | Open the stand mixer head. |
| Atomic | `PickPlaceCounterTo Cabinet` | 1 | No | Pick the item from the counter and place it in the cabinet. |
| Atomic | `PickPlaceCounterTo Stove` | 1 | No | Pick the item from the plate and place it in the pan. |
| Atomic | `PickPlaceDrawerTo Counter` | 1 | No | Pick the item from the drawer and place it on the counter. |
| Atomic | `PickPlaceSinkTo Counter` | 1 | No | Pick the item from the sink and place it on the container located on the counter. |
| Atomic | `PickPlaceToasterTo Counter` | 1 | No | Place the toasted item on a plate. |
| Atomic | `SlideDishwasherRack` | 1 | No | Fully slide the top dishwasher rack [*in/out*]. |
| Atomic | `TurnOffStove` | 1 | No | Turn off the [*burner location*] burner of the stove. |
| Atomic | `TurnOnElectricKettle` | 1 | No | Press down the lever to turn on the electric kettle. |
| Atomic | `TurnOnMicrowave` | 1 | No | Press the start button on the microwave. |
| Atomic | `TurnOnSinkFaucet` | 1 | No | Turn on the sink faucet. |

Figure 7: **Post-training Atomic-Seen Tasks (18)**

Note: The "MoMa req." column indicates whether the task requires Mobile Manipulation or base navigation.

| Activity | Task | # Sub-tasks | MoMa req. | Description |
|---|---|---|---|---|
| Serving beverages | DeliverStraw | 4 | Yes | Take a straw from the drawer in front and place it inside the glass cup on the dining counter. |
| Toasting bread | GetToastedBread | 4 | Yes | Start the toaster. Once the lever pops up, take the bread to the plate on the dining counter. |
| Brewing | KettleBoiling | 2 | No | Pick the kettle from the counter and place it on a stove burner. Then turn the burner on. |
| Loading dishwasher | LoadDishwasher | 3 | No | Pick up the items from the counter, place them in the dishwasher, and close the dishwasher door. |
| Packing lunches | PackIdentical Lunches | 15 | Yes | Place two identical items of each object in each tupperware on the nearby counter, to pack two identical lunches. |
| Washing dishes | PreSoakPan | 3 | No | Pick the pan and sponge and place them into the sink. Then turn on the water. |
| Brewing | PrepareCoffee | 2 | No | Pick the mug from the cabinet, place it under the coffee machine dispenser, and press the start button. |
| Cleaning sink | RinseSinkBasin | 2 | No | Turn on the sink and manuever the spout to wash all locations of the sink basin. |
| Sanitizing cutting boards | ScrubCutting Board | 2 | Yes | Pick up the sponge from the counter and clean the cutting board by briefly scrubbing or pressing down on the cutting board. Once finished, release the sponge. |
| Frying | SearingMeat | 3 | Yes | Grab the pan from the cabinet and place it on the [*burner location*] burner on the stove. Then place the item on the stove and turn the burner on. |
| Slicing meat | SetUpCutting Station | 2 | Yes | Pick up the knife from the drawer and place it on the cutting board. Then place the meat from the plate to the cutting board. |
| Organizing dishes and containers | StackBowls Cabinet | 2 | Yes | Pick up the bowls on the counter and stack them on top of one another in the open cabinet. Place the smaller bowl on top of the larger bowl. |
| Steaming food | SteamIn Microwave | 6 | Yes | Pick the item from the sink and place it in the bowl. Then pick the bowl and place it in the microwave. Then close the microwave door and press the start button. |
| Sauteing vegetables | StirVegetables | 4 | Yes | Put the items in the pot. Retrieve the spatula and lightly stir the vegetables in the pot. |
| Storing leftovers | StoreLeftovers InBowl | 5 | Yes | Pick the chicken drumstick and item from their plates and place them in the bowl. Then put the bowl in the fridge. |
| Making salads | WashLettuce | 2 | No | Wash the lettuce in the sink by running water over it. |

Figure 8: **Post-training Composite-Seen Tasks (16)**

| Activity | Task | # Sub-tasks | MoMa req. | Description |
|---|---|---|---|---|
| Setting the table | `ArrangeBread Basket` | 5 | Yes | Open the cabinet, pick up the item from the cabinet and place it in the basket. Then move the basket to the dining counter. |
| Brewing | `ArrangeTea` | 3 | No | Pick the kettle from the counter and place it on the tray. Then pick the mug from the cabinet and place it on the tray. Then close the cabinet doors. |
| Making toast | `Bread Selection` | 2 | Yes | From the different types of pastries on the counter, select a croissant and place it on the cutting board. Then retrieve a jar of jam from the cabinet and place it alongside the croissant on the cutting board. |
| Arranging condiments | `Categorize Condiments` | 2 | No | Put the shaker and condiment bottle from the counter next to their counterparts in the cabinet. |
| Chopping vegetables | `CuttingTool Selection` | 2 | No | Place the appropriate cutting tool for cutting the item skin on the cutting board. |
| Garnishing dishes | `Garnish Pancake` | 4 | Yes | Take the strawberry from the fridge and place it on top of the pancake, located on the dining counter. |
| Arranging cabinets | `Gather Tableware` | 4 | Yes | Gather all objects into one cabinet and sort the glasses and bowls to opposite sides. |
| Preparing sandwiches | `HeatKebab Sandwich` | 6 | Yes | Pick up the kebab skewer and baguette bread, and place them inside the toaster oven. Close the toaster oven door and start by setting the timer. |
| Adding ice to beverages | `MakeIce Lemonade` | 5 | Yes | Grab a lemon wedge from the fridge and one ice cube from the ice bowl, and put them in the glass of lemonade. |
| Serving food | `PanTransfer` | 3 | No | Pick up the pan and dump the vegetables in it onto the plate. Then return the pan to the stove. |
| Portioning meals | `PortionHot Dogs` | 4 | Yes | Place one bun and one sausage from the bowl on each plate. |
| Organizing recycling | `Recycle BottlesBy Type` | 3 | Yes | Move the plastic bottles in the middle to the plastics group, and the glass bottles in the middle to the glass group. |
| Managing freezer space | `Separate FreezerRack` | 7 | Yes | Take the meat container that has the meat item(s) and place it on the second highest rack of the freezer. Then take the vegetable container that has the vegetable(s) and place it on the highest rack of the freezer. |
| Reheating food | `WaffleReheat` | 4 | Yes | Open the microwave, place the bowl with waffle inside the microwave, then close the microwave door and turn it on. |
| Washing produce | `WashFruit Colander` | 4 | No | Put the colander in the sink, put the item in the colander, and turn on the sink faucet and pour water over the colander. |
| Measuring ingredients | `Weigh Ingredients` | 2 | No | Pick the item and place it on the digital scale for weighing, and close the cabinet. |

Figure 9: **Post-training Composite-Unseen Tasks (16)**

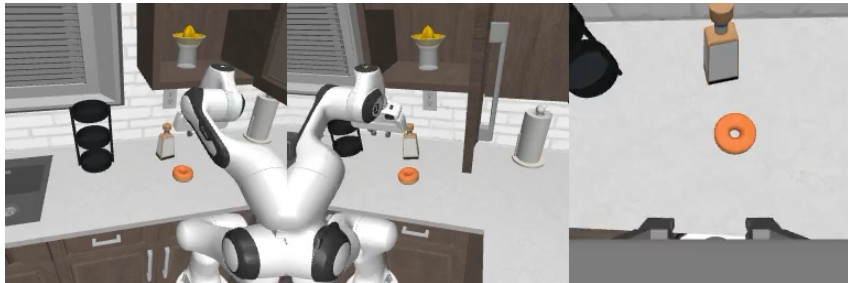

Figure 10: **Camera Images.** Camera images from three views (rendered at $256 \times 256$ resolution are fed into the model.)

## F    DATASETS

We present an overview of our datasets in Table 7.

| Setting | Num Tasks | Num Scenes | Demos per Task | Dataset Size (hrs) |
|---|---|---|---|---|
| Pretraining (Human) | 300 | 2500 | 100 | 404 |
| Pretraining (MimicGen) | 60 | 2500 | 10,000 | 1615 |
| Target (Human) | 50 | 10 | 500 | 208 |

Table 7: Dataset statistics across pretraining and target settings.

For each demonstration, we store the language instruction, proprioceptive information (robot base pose, robot end effector pose, gripper state information), images from three cameras (wrist camera, left third-person camera, right third-person camera), and the actions.

## G    POLICY LEARNING

### G.1    MODEL ARCHITECTURES AND TRAINING PROTOCOL

Our experiments focus on training vision-based models. The model takes as input a combination of low-level proprioceptive information (base pose, end effector pose, gripper state), task instruction language, and camera images (one wrist camera image and two third-person camera images).

Each of the camera images is at $256 \times 256$ resolution, and we show examples of the camera views in Figure 10.

We experiment with four models:

**Diffusion Policy.** Diffusion Policy models trajectory generation as a conditional denoising process in action space, recovering actions from noisy expert trajectories to handle multi-modal robot behaviors. We use an open source diffusion policy codebase and add language conditioning by fusing CLIP-based language embeddings (Radford et al., 2021) with the ResNet visual encoder via FiLM conditioning layers (Perez et al., 2018). We use the transformer diffusion variant, with a 12-layer transformer with an embedding dimension of $512$. We train the model with a batch size of 192 and train for 250k steps for the multi-task learning experiment.

$\pi_0 / \pi_{0.5}$. $\pi_0$ and $\pi_{0.5}$ are vision-language-action models which use PaLI Gemma (Beyer et al., 2024) as the underlying VLM and fuses an action expert to output robot actions via flow matching (Lipman et al., 2023). We use the official open source repository. We use the default full fine-tuning configuration, and we use a batch size of $64$ (the highest batch size we can fit on a GH200 GPU). For the multi-task learning experiment, we train the model for 75k steps (48 hours of training time on a GH200 GPU).

**GR00T N1.5**. GR00T N1.5 is a vision-language-action model which uses a system1-system2 architecture, with the Eagle2 VLM (Li et al., 2025) serving as the high-level encoder (system2) and

an action decoder to produce actions via flow matching (system1). We use the official open source repository. For all experiments, we freeze the vision encoder and language encoder (which are the default settings from the open source codebase), and we use a batch size of $128$ (the highest batch size we can fit on a GH200 GPU). For the multi-task learning experiments, we train for 120k steps. For the foundation model training experiments, we pretrain for 80k steps and fine-tune on target data for 60k steps. Finally, for lifelong learning experiments, we train stage1 for 100k steps, followed by 60k steps for all subsequent stages of training. Generally, we find these settings to be sufficient to allow for model convergence.

## G.2 Evaluation Protocol

After training, we evaluate the model at a specified checkpoint on a suite of evaluation tasks. For each evaluation task, we run 30 trials for a specified maximum number of timesteps (the maximum duration is task-dependent). If during this duration the agent achieves the task success condition (binary condition), the episode is counted as a success; otherwise, a failure. We report the average success rate across tasks.

## H Additional Experiments

### H.1 Pretraining Scene Diversity

Our pretraining data spans 2,500 kitchen scenes (50 layouts $\times$ 50 styles), and we compare to restricting pretraining data to 25 scenes (5 layouts $\times$ 5 styles), and 5 scenes (5 layouts $\times$ 1 style). To run a fair comparison across these settings, we use MimicGen to generate demonstrations for each setting, generating data across 17 atomic tasks in pretraining kitchens. We run zero-shot evaluations on the 10 fixed target kitchen scenes, and also try learning on the atomic target data with 50 demonstrations per task. See Table 8 for results. For zero-shot evaluation, we observe notable performance gains as the number of pretraining scenes increases. These gains also hold in subsequent target task fine-tuning, highlighting the need for diverse pretraining data.

|  | Pretraining Data | | |
|---|---|---|---|
|  | 5 Scenes | 25 Scenes | 2500 Scenes |
| Zero-shot Evaluation | 29.6 | 39.6 | 44.7 |
| + Fine-tuning on Atomic Target Data (10%) | 53.3 | 56.7 | 62.4 |

Table 8: **Pretraining scene diversity results.** Increasing the composition of scenes in pretraining data improves downstream task performance.

### H.2 Robustness Evaluations

In order to examine the generalization capabilities endowed by pretraining on our data, we perform a set of robustness evaluations on the GR00T N1.5 model trained on the full pretraining and target mixture. We perturb an aspect of the model's input and evaluate it on our `Composite-Seen` and `Composite-Unseen` tasks. Specifically, we look at the following perturbations:

- **Novel Language**: We prompt an LLM for novel but semantically similar task instructions.
- **Novel Joint Angles**: We sample Gaussian noise and add it to the starting joint angles of the robot.
- **Novel Base Pose**: We sample Gaussian noise and add it to the starting position and yaw of the robot base.
- **Novel Camera Pose**: We sample Gaussian noise and add it to the default third-person and wrist camera poses.

We find that the model is robust to language variations, but can suffer from novel camera poses, joint angles, and base poses.

Table 9: Evaluation of robustness under different perturbations.

| Task Split | No Perturbation | Novel Language | Camera Perturbations | Initial Joint Noise | Initial Base Pose Noise |
|---|---|---|---|---|---|
| Composite-Seen | 40.6 | 38.3 | 28.8 | 27.9 | 31.2 |
| Composite-Unseen | 42.1 | 39.2 | 31.5 | 32.1 | 30.2 |

### H.3 Joint Co-Training of Pretraining and Target Data

As an extension to the foundation model training experiments, we examine a separate variant in which we train on all pre-training data and 100% of the target data jointly in one single phase. We trained the model for 120k steps and report the resulting task success rates in the target kitchens as follows:

- `Atomic-Seen`: 44.1%
- `Composite-Seen`: 9.0%
- `Composite-Unseen`: 11.7%
- Average: 22.5%

Compared to our two-stage learning framework (pretraining first, followed by fine-tuning on target data), performance under this co-training regime is substantially lower. This result underscores the importance of a dedicated fine-tuning phase for learning highly performant policies tailored to the target tasks.

### H.4 LoRA Fine-tuning

For the multi-task learning experiments in section 4.2, we ran GR00T N1.5 with LoRA fine-tuning, trained for the same number of steps, batch size, etc, as the full fine-tuning variant. The results are as follows:

Table 10: Policy success rates (%) comparing full vs LoRA fine-tuning for GR00T N1.5.

| | Atomic-Seen | Composite-Seen | Composite-Unseen | **Average** |
|---|---|---|---|---|
| **Full fine-tuning** | 43.0 | 9.6 | 4.4 | 20.0 |
| **LoRA fine-tuning** | 2.4 | 0.2 | 0.8 | 1.2 |

Full fine-tuning is critical to model performance.

## I Additional Analysis

### I.1 Foundation Model Training Analysis

We break down the per-task task performance for the best performing variant, pretraining followed by fine-tuning on target data (on 100% of data), in Table 11. Among the `Atomic-Seen` tasks, the worst-performing tasks are `TurnOffStove` and `CloseBlenderLid`, which involve high precision and dexterity. However, for `Composite-Seen` and `Composite-Unseen` tasks, the worst-performing tasks span many diverse characteristics. We run a qualitative analysis, outlining common failure modes for the tasks where the model performs at a 30% or less success rate:

- `SteamInMicrowave`: difficulty placing the bowl in the microwave, either placing on the edge of the microwave or dropping the bowl in the air right before placing it in the microwave
- `SearingMeat`: typically does not turn on the stove burner correctly, or attempts to turn on the incorrect stove burner; or does not place the pan on a valid location on the stovetop
- `PackIdenticalLunches`: unreliable picking from fridge; not moving to the tupperware to place items; placing items in wrong tupperware
- `PrepareCoffee`: often does not place the coffee mug correctly under the coffee machine

- `DeliverStraw`: range of failures: difficulty opening drawers, difficulty transporting straw (dropping it), difficulty placing straw into cup

- `GetToastedBread`: often does not press down on lever fully; sometimes presses down lever but then acts randomly

- `PortionHotDogs`: unreliable picks from crowded bowl; unreliable place by placing item on counter instead of plate; placing items on the wrong plate

- `PanTransfer`: generally picks up the pan but does not reliably flip the contents of the pan into the plate; this is a dynamic task that's quite unique, does not have much overlap with other tasks in the benchmark

- `HeatKebabSandwich`: often fails to pull out the toaster rack; other times often after placing the first item on the rack inadvertently pushes the rack in by accident and does not place the second item in

- `CategorizeCondiments`: pick and place is not reliable, or does not place matching condiments next to each other

- `SeparateFreezerRack`: often fails to reliably place the tupperware into the freezer, as the freezer is a tight space

- `GatherTableware`: must locate the other mug from the kitchen, and bring it back; the navigation ability here is not reliable; also sometimes does not place the mug inside the cabinet, drops it in the air without reaching far into the cabinet.

| Task | Success Rate (%) | Stages | MoMa Required |
|------|------------------|--------|---------------|
| Atomic-Seen | | | |
| TurnOnElectricKettle | 93 | 1 | No |
| OpenStandMixerHead | 90 | 1 | No |
| CloseToasterOvenDoor | 87 | 1 | No |
| OpenCabinet | 87 | 1 | No |
| SlideDishwasherRack | 87 | 1 | No |
| PickPlaceToasterToCounter | 73 | 1 | No |
| TurnOnMicrowave | 70 | 1 | No |
| OpenDrawer | 70 | 1 | No |
| PickPlaceSinkToCounter | 70 | 1 | No |
| PickPlaceCounterToStove | 70 | 1 | No |
| CloseFridge | 67 | 1 | No |
| TurnOnSinkFaucet | 63 | 1 | No |
| PickPlaceCounterToCabinet | 63 | 1 | No |
| CoffeeSetupMug | 60 | 1 | No |
| NavigateKitchen | 60 | 1 | Yes |
| PickPlaceDrawerToCounter | 50 | 1 | No |
| TurnOffStove | 37 | 1 | No |
| CloseBlenderLid | 37 | 1 | No |
| Composite-Seen | | | |
| StackBowlsCabinet | 83 | 2 | Yes |
| PreSoakPan | 70 | 3 | No |
| ScrubCuttingBoard | 70 | 2 | Yes |
| WashLettuce | 67 | 2 | No |
| RinseSinkBasin | 60 | 2 | No |
| KettleBoiling | 53 | 2 | No |
| LoadDishwasher | 47 | 3 | No |
| StoreLeftoversInBowl | 43 | 5 | Yes |
| SetUpCuttingStation | 33 | 2 | Yes |
| StirVegetables | 33 | 4 | Yes |
| SteamInMicrowave | 30 | 6 | Yes |
| SearingMeat | 27 | 3 | Yes |
| PackIdenticalLunches | 17 | 15 | Yes |
| PrepareCoffee | 13 | 2 | No |
| DeliverStraw | 3 | 4 | Yes |
| GetToastedBread | 0 | 4 | Yes |
| Composite-Unseen | | | |
| RecycleBottlesByType | 87 | 3 | Yes |
| WaffleReheat | 83 | 4 | Yes |
| ArrangeBreadBasket | 77 | 5 | Yes |
| WeighIngredients | 67 | 2 | No |
| BreadSelection | 60 | 2 | Yes |
| CuttingToolSelection | 47 | 2 | No |
| GarnishPancake | 47 | 4 | Yes |
| ArrangeTea | 43 | 3 | No |
| WashFruitColander | 40 | 4 | No |
| MakeIceLemonade | 40 | 5 | Yes |
| PortionHotDogs | 23 | 4 | Yes |
| PanTransfer | 20 | 3 | No |
| HeatKebabSandwich | 13 | 6 | Yes |
| CategorizeCondiments | 10 | 2 | No |
| SeparateFreezerRack | 10 | 7 | Yes |
| GatherTableware | 7 | 4 | Yes |

Table 11: **Foundation Model Training Results.**

