# OpenReview forum: "RoboCasa365: A Large-Scale Simulation Framework for Training and Benchmarking Generalist Robots"
_ICLR.cc/2026/Conference — ICLR 2026 Poster_

### Official Review · Reviewer_S6tn · 2025-10-16

**Soundness:** 4
**Presentation:** 4
**Contribution:** 4
**Rating:** 10
**Confidence:** 5

**Summary:**

This paper introduces RoboCasa365, a large-scale simulation framework designed for training and benchmarking generalist robot policies for everyday household tasks. Building upon the existing RoboCasa platform, the authors make significant contributions by massively scaling up the content and creating a structured evaluation suite. The framework includes 2,500 diverse kitchen environments derived from real-world home layouts, a comprehensive set of 365 everyday tasks (65 atomic and 300 composite), and over 2,200 hours of robot interaction data (both human-collected and synthetically generated via MimicGen). The work establishes benchmarks for three critical research areas: massively multi-task training, foundation model training (pre-training and fine-tuning), and lifelong learning. Through extensive experiments, the authors evaluate several state-of-the-art models (Diffusion Policy, π₀, and GR00T N1.5), demonstrating the utility of the benchmark for analyzing the impact of data scale, task diversity, and pre-training on policy generalization.

**Strengths:**

- The primary strength of this work is the sheer scale and diversity of the simulation environment and accompanying datasets. The introduction of 2,500 unique kitchen scenes modeled after real-world homes and 365 distinct tasks represents a monumental step up from prior simulation benchmarks, which are often limited to a few scenes or a narrow set of tasks. This scale is critical for training and rigorously evaluating the generalization capabilities of high-capacity foundation models.
- The authors have thoughtfully structured the entire framework to facilitate rigorous scientific inquiry. The explicit separation of assets, scenes, and tasks into "pre-training" and "post-training" splits  is a key design choice that enables controlled studies on generalization, data efficiency, and transfer learning, which are central questions in the field.
- The paper provides not just a dataset, but a full suite of defined benchmarks for multi-task learning, foundation model training, and lifelong learning. This provides a ready-to-use toolkit for researchers to compare methods reproducibly and systematically analyze model performance across different learning paradigms.
- The significant expansion of interactable and articulated appliances (from 20 to 456 instances across 12 categories)  is a substantial contribution. This allows for the creation of more complex, realistic, and long-horizon tasks  that better reflect the challenges of real-world household robotics, moving beyond simple pick-and-place scenarios.

**Weaknesses:**

While acknowledging the sim-to-real gap as a limitation, the paper lacks any empirical grounding or discussion of how its simulation-based success metrics might correlate with real-world performance. This is a critical omission for a benchmark intended to gauge progress towards deployable generalist robots. Recent work such as STAR-Gen[1]  proposes a detailed taxonomy for evaluating real-world generalization across distinct visual, semantic, and behavioral axes. The binary task success metric used in RoboCasa365 is coarse and may not be predictive of a policy's performance on these fine-grained real-world challenges. For instance, a policy might succeed across many simulated scenes but still be brittle to specific real-world semantic or behavioral perturbations not adequately modeled in simulation (e.g., changes in object mass or nuanced language instructions).

[1] Gao et al. A Taxonomy for Evaluating Generalist Robot Policies.

**Questions:**

The multi-task evaluation includes GR00T N1.5, a model primarily developed for humanoid robots. While its strong performance is interesting, could you discuss the implications of evaluating a humanoid-centric foundation model on a mobile manipulator task set? Does this suggest a high degree of embodiment-agnostic knowledge transfer, or are there specific architectural features (e.g., in its vision-language processing) that are particularly well-suited to the benchmark's challenges, independent of the final embodiment?

---

> ### Author Response · Authors · 2025-11-24
> **Author Response to Reviewer S6tn**
>
> Thank you very much for your detailed and thoughtful review! We really appreciate your enthusiasm and strong endorsement of our work. We address your comments below:
>
> > While acknowledging the sim-to-real gap as a limitation, the paper lacks any empirical grounding or discussion of how its simulation-based success metrics might correlate with real-world performance. This is a critical omission for a benchmark intended to gauge progress towards deployable generalist robots.
>
> Thank you for the feedback. Following your suggestion, we conducted a set of real-world experiments. We assembled a full robotic stack using the DROID Panda arm in a real kitchen environment and collected demonstrations for four tasks:
> - CloseElectricKettleLid: Close the lid of an electric kettle.
> - PickPlaceToasterOvenToCounter: Move an item from the toaster oven to the counter.
> - PickPlaceCounterToCabinet: Move an object from the counter into a cabinet.
> - PlaceOnDishRack: A longer-horizon task involving placing two items from the sink onto a dish rack.
>
> We collected 30 demonstrations for each of the first three tasks and 50 demonstrations for the final task, yielding 140 real-world demonstrations in total. We then compared two training settings:
> - Real-Only: Training GR00T exclusively on the 140 real-world demonstrations.
> - Sim-and-Real (ours): Pre-training GR00T on our simulation tasks (using data from the top half of best-performing simulation tasks), followed by co-fine-tuning on the real-world demonstrations along with analogous simulated data for the four real-world tasks.
>
> After training, we evaluated each model in the real world, running 20 trials per task and reporting success rates. The Real-Only model achieved a mean success rate of 61.8%, whereas our Sim-and-Real approach achieved 79.8%, a substantial improvement. These results highlight the value of our simulation benchmark, not only as a platform for studying and comparing algorithms in simulation, but also as an effective tool for improving real-world policy learning.
>
> | Task                         | CloseElectricKettleLid | PickPlaceToasterOvenToCounter | PickPlaceCounterToCabinet | PlaceOnDishRack | Average |
> |------------------------------|-------------------------|-------------------------------|----------------------------|------------------|---------|
> | **Real-Only**                | 70%                    | 70%                           | 52%                        | 55%              | 61.8%   |
> | **Sim-and-Real (ours)**      | 70%                    | 100%                          | 84%                        | 65%              | 79.8%   |
>
>
> > Recent work such as STAR-Gen[1] proposes a detailed taxonomy for evaluating real-world generalization across distinct visual, semantic, and behavioral axes. The binary task success metric used in RoboCasa365 is coarse and may not be predictive of a policy's performance on these fine-grained real-world challenges. For instance, a policy might succeed across many simulated scenes but still be brittle to specific real-world semantic or behavioral perturbations not adequately modeled in simulation (e.g., changes in object mass or nuanced language instructions).
>
> Thank you for raising this point. We agree that analyzing performance across different axes—visual, semantic, and behavioral—can provide valuable insights. In the initial submission, Appendix G.2 includes experiments assessing the robustness of the trained model across several axes, including language instructions, robot initialization, and camera pose. We found that while the model is relatively robust to novel language instructions, it can struggle to generalize to novel camera poses or robot initializations. We hope these results are useful for practitioners aiming to develop robust models across these axes.
>
> > The multi-task evaluation includes GR00T N1.5, a model primarily developed for humanoid robots. While its strong performance is interesting, could you discuss the implications of evaluating a humanoid-centric foundation model on a mobile manipulator task set?
>
> This is a very insightful point! We would like to highlight that, while GR00T N1.5 is primarily presented as a foundation model for humanoids, the training mix used to develop the underlying base GR00T N1 model includes cross-embodied data, with a substantial portion coming from single-arm robots. One representative pre-training dataset is DROID, which features the same robot arm (Franka Emika Panda) used in our experiments. Understanding the impact of embodiment on downstream performance in our benchmark is an interesting direction. One way to explore this would be to conduct systematic comparisons across the pre-training datasets used for the base GR00T model, which we leave to the broader research community to investigate.

---

### Official Review · Reviewer_tQ4v · 2025-10-30

**Soundness:** 3
**Presentation:** 3
**Contribution:** 3
**Rating:** 6
**Confidence:** 4

**Summary:**

The authors presented RoboCasa365, a large-scale simulation framework for benchmarking generalist robot policies, and offer large pre-training and post-training datasets as well. It provides 2500 kitchen environments with 2000 hours of simulated robot data.

**Strengths:**

- The authors provide a valuable resource for the community to benchmark generalist robot policies: there is really high diversity in simulated scenes, and a large amount of demonstrations for pre-training and post-training
- There are a large number (365) of tasks spanning across 50 different categories. There is a large number of scenes, with 50 different layouts and each have 50 different styles. This is a huge improvement over prior work.
- The authors also provide a good tool to benchmark lifelong learning, and the dataset includes tasks with various numbers of subtasks.
- The authors provide good experiments to demonstrate the use of the benchmark and the dataset

**Weaknesses:**

Overall this is a good paper and a great resource for the community. However, I think the experiments section is still a bit lacking, and misses a good opportunity to provide more insights into the dataset collected by the authors.

- The writing in the experiments section is somewhat unclear: (1) It is unclear what the evaluation protocol is and how different models are compared (aka do you report success rate? Are there partial success? etc.). It is also unclear what the numbers mean in Table 1: is it success rate? (2) There lack discussion in the main text on the difference between the three models being compared, and lacks discussion on why GR00T N1.5 may be performing the best.  (3) Table 2 doesn’t bold the best numbers
- In section 4.2 (specifically Figure 5 and Table 2), in would be interesting to see comparison with: finetune GR00NT 1.5 public checkpoint on a mix of pre-training + post-training data, but does not first do pre-training and then do fine-tuning, and instead just do “pre-training” over the entire data mix.
- It is unclear what checkpoints the authors chose to evaluate for the experiments, and how they chose it. It is unclear how long “pre-training” is and how many epochs over the dataset it’s taken.
- In Section 4.4, the authors observed that Human300 outperforms Human300+MG60, and they hypothesized that it is because “adding MimicGen data dilutes the contribution of other human datasets”. However, there is not support for this hypothesis. One experiment to verify this hypothesis would be to downweight the MG60 data in pre-training and compare. This is very important as a huge chunk of the dataset is composed of MimicGen data, and it would be important to know how useful this data is (or whether it is useful at all). If it is not useful, then the actual dataset would be much less than the 2000h claimed in the paper, and would need to adjust the writing to reflect that.
- Related work section lacks full discussion on benchmarks for generalist robots outside of simulation, such as [1][2][3] and live competitions like [4] [5]

[1] Train offline, test online: A real robot learning benchmark
[2] Homerobot: Open-vocabulary mobile manipulation
[3] AutoEval: Autonomous Evaluation of Generalist Robot Manipulation Policies in the Real World
[4] The DARPA robotics challenge finals: Results and perspectives
[5] Analysis and observations from the first amazon picking challenge

**Questions:**

See my main points in the weakness section.
Other small questions:
- There is no discussion of the underlying simulation framework and how it compares to other simulation frameworks: how fast is it? Is GPU rendering supported?
- Can you say more about how the MG60 dataset is generated and how diverse it is?
- How optimal are the human teleoperated demonstration data?

---

> ### Author Response · Authors · 2025-11-24
> **Author Response to Reviewer tQ4v (Part 1/2)**
>
> Thank you very much for your detailed and thoughtful review! We appreciate your positive comments and constructive feedback. We address your comments below:
>
> > It is unclear what the evaluation protocol is and how different models are compared (aka do you report success rate? Are there partial success? etc.). It is also unclear what the numbers mean in Table 1: is it success rate?
>
> For all experiments in the paper, we report the task success rate. The task success rate is a sparse signal (either 1 if the task is fully successfully completed or 0 otherwise). We do not report partial success scores, as many tasks can be completed in multiple valid ways, making it challenging to design a consistent and scalable partial-success criterion across all tasks. We have updated the manuscript to clarify our evaluation metrics.
>
> > There lack discussion in the main text on the difference between the three models being compared,
>
> In order to address the space limitations, we have included more details in Appendix F.1, where we outline the methods, architectures, hyperparameters, and training protocols.
>
> > lacks discussion on why GR00T N1.5 may be performing the best.
>
> While our multi-task learning experiments show that GR00T outperforms the Diffusion Policy and pi0 baselines, we do not claim that GR00T is definitively the superior method. Performance can be influenced by many factors, including the amount of compute used (e.g., batch size), hyperparameters such as learning rate, and whether the visual or language backbones are fine-tuned.
> One possible explanation for the observed gap is that we use a larger batch size for GR00T (128) than for pi0 (64), made possible by GR00T’s smaller memory footprint (GR00T, by default, does not fine-tune the visual encoder, whereas pi0 does). We adopt the default hyperparameter settings for both GR00T and pi0 and use the largest batch size that fits on a single GH200 GPU. It is entirely plausible that with different hyperparameters or greater compute resources, pi0 could match or exceed GR00T’s performance.
>
> > Table 2 doesn’t bold the best numbers
>
> We have bolded the best performing variant, which is pretraining followed by post-training on 100% of the post-training data.
>
> > In section 4.2 (specifically Figure 5 and Table 2), in would be interesting to see comparison with: finetune GR00NT 1.5 public checkpoint on a mix of pre-training + post-training data, but does not first do pre-training and then do fine-tuning, and instead just do “pre-training” over the entire data mix.
>
> Thank you for the suggestion. We conducted the proposed experiment, in which we train on all pre-training data and 100% of the post-training data simultaneously. We trained the model for 120k steps and report the resulting task success rates in the post-training kitchens:
>
> - Atomic-Seen: 44.1%
> - Composite-Seen: 9.0%
> - Composite-Unseen: 11.7%
>
> Compared to our two-stage learning framework (pre-train followed by post-train), performance under this co-training regime is substantially lower. This result underscores the importance of a dedicated post-training phase for learning highly performant policies tailored to the post-training tasks. We have added these results in Appendix G.3.
>
> > It is unclear what checkpoints the authors chose to evaluate for the experiments, and how they chose it. It is unclear how long “pre-training” is and how many epochs over the dataset it’s taken.
>
> For all of our foundation model training experiments with GR00T, models are pre-trained for 80k steps and post-trained for 60k steps, which we found sufficient for task success rates to generally converge. We discuss these details in Appendix F.1. While larger batch sizes and longer training schedules could potentially further improve performance, we limited our experiments due to compute resource constraints.

---

> > ### Author Response · Authors · 2025-11-24
> > **Author Response to Reviewer tQ4v (Part 2/2)**
> >
> > > In Section 4.4, the authors observed that Human300 outperforms Human300+MG60, and they hypothesized that it is because “adding MimicGen data dilutes the contribution of other human datasets”. However, there is not support for this hypothesis.
> >
> > Indeed, our experiments do not currently show strong positive benefits from incorporating MimicGen data during pre-training improves performance. Although MimicGen allows us to generate large quantities of synthetic trajectories at scale, we find that these are mixed quality demonstrations. We have included examples of MimicGen demonstrations in the supplementary materials for reference. Such mixed quality data can yield suboptimal performance when using standard imitation learning algorithms.
> >
> > Our work does not make a definitive statement on the value of synthetic demonstration data. We believe that the utility of large-scale synthetic data is dependent on the training algorithm and policy architecture, and our work only explores a small set of current policy learning methods. By releasing this dataset, we hope to enable future work that can more effectively leverage large-scale data of mixed quality and further explore how synthetic demonstrations can complement human demonstrations.
> >
> > > Related work section lacks full discussion on benchmarks for generalist robots outside of simulation
> >
> > Thank you for the suggestions! We’ve added the suggested references to the manuscript.
> >
> > > There is no discussion of the underlying simulation framework and how it compares to other simulation frameworks: how fast is it? Is GPU rendering supported?
> >
> > RoboCasa365 is built on top of RoboSuite, which uses the MuJoCo physics engine. While MuJoCo’s core physics computations are CPU-based, we leverage GPU-based rendering. RoboCasa365 simulates at 20 Hz, with the simulation running approximately in real time—slightly faster or slower depending on scene complexity and hardware specifications. Multiple asynchronous environments can be run in parallel, allowing overall throughput to scale with the number of available CPU cores and GPUs. We have added this discussion to Appendix I.1.
> >
> > > Can you say more about how the MG60 dataset is generated and how diverse it is?
> >
> > We use the open-source MimicGen codebase released by RoboCasa (Nasiriany et al., 2024), which supports 24 atomic tasks, and extend it to cover a total of 60 atomic tasks. For each task, we define object-centric subtasks. For example, the CloseBlenderLid task includes two stages: (1) the robot reaches the lid, and (2) the robot places the lid on the counter.
> > For each of the 60 MimicGen-supported tasks, we start with 100 human source demonstrations and generate 10,000 synthetic demonstrations. These 10,000 demonstrations are distributed across 2,500 kitchen scenes. Example videos illustrating the diversity and quality of these demonstrations are provided in the supplementary materials.
> >
> > > How optimal are the human teleoperated demonstration data?
> >
> > Quantifying the optimality of teleoperation data is challenging, but we take steps to ensure it is near-optimal. We discard episodes that involve dropped objects, jerky robot motions, excessive grasping attempts, or collisions with the environment. The supplementary material (zip file in the submission) has been updated to include videos of the teleoperation demonstrations for reference.

---

> > > ### Comment · Reviewer_tQ4v · 2025-11-26
> > >
> > > Thanks to the authors for providing a detailed response, that answers most of my questions. My only and major concern now is still regarding MG60. I understand that the paper is not trying to make a statement on the value of synthetic data, but I think the current writing obfuscates this fact. As the writing is currently, it appears to the readers that there are 2200+ hours of "same quality" data, but in fact a lot of them *may* be of inferior quality. I would strongly recommend the authors to be more upfront and nuanced about this, and state out right in the abstract, introduction, and Figure 1 that there is 600+ hours of human demonstration data and 1600+ hours of synthetic MG data.

---

> > > > ### Author Response · Authors · 2025-12-03
> > > > **Manuscript updated**
> > > >
> > > > Thank you for your feedback! We have uploaded a new version of the manuscript to clarify further the distinction between human and synthetic data. This is reflected in the abstract, Figure 1, introduction, and section 4.4.

---

### Official Review · Reviewer_jCDL · 2025-10-30

**Soundness:** 3
**Presentation:** 3
**Contribution:** 3
**Rating:** 6
**Confidence:** 3

**Summary:**

The authors propose a large scale imitation learning benchmark for household manipulation tasks, for studying pretraining and postraining of large scale generalist policy architectures. There are 2500 distinct kitchens, and 365 (some mobile) manipulation tasks, ranging from atomic to compositional.  The authors collect 100 demos for each pretraining task (300 out of 365), and also use the MimicGen framework to generate an additional 10k synthetic demonstrations for each task. The authors evaluate generalist policies in zero-shot task generalization after pretraining, improvement from pretraining and posttraining, and lifelong learning. They find that GROOT is the best for generalization, lifelong learning is still challenging, and that pretraining data composition matters over pure quantity.

**Strengths:**

The benchmark seems quite comprehensive and the experiments are sensible and  systematic.

One very interesting result is that synthetic demos do not help at all. In the pretraining data study, they show that just using the human demos is good enough. I would like to see additional explanation in why MimicGen is not helpful, since MimicGen itself reports positive gains in using synthetic demos.

The paper is easy to read.

**Weaknesses:**

This is not really a substantial weakness, but the experiments and results are very straightforward and almost boring. It would be nice to see more interesting or qualitative phenomena, such as why synthetic demos are harmful, why GROOT outperforms other generalist policies (i.e. is hierarchy helpful), seeing if LoRA finetuning affects performance, both in 4.2 and 4.3, etc. Another suggestion is to stratify the tasks, like contact rich tasks, mobile manipulation tasks, etc. and see performance profiles across different tasks.

**Questions:**

See weaknesses and questions above.

---

> ### Author Response · Authors · 2025-11-24
> **Author Response to Reviewer jCDL**
>
> Thank you very much for taking the time to review our paper! We address your comments as follows:
>
> > why synthetic demos are harmful
>
> Indeed, our experiments do not currently show strong positive benefits from incorporating MimicGen data during pre-training improves performance. Although MimicGen allows us to generate large quantities of synthetic trajectories at scale, we find that these are mixed quality demonstrations. We have included examples of MimicGen demonstrations in the supplementary materials for reference. Such mixed quality data can yield suboptimal performance when using standard imitation learning algorithms.
>
> Our work does not make a definitive statement on the value of synthetic demonstration data. We believe that the utility of large-scale synthetic data is dependent on the training algorithm and policy architecture, and our work only explores a small set of current policy learning methods. By releasing this dataset, we hope to enable future work that can more effectively leverage large-scale data of mixed quality and further explore how synthetic demonstrations can complement human demonstrations.
>
> > why GROOT outperforms other generalist policies (i.e. is hierarchy helpful)
>
> While our multi-task learning experiments show that GR00T outperforms the Diffusion Policy and pi0 baselines, we do not claim that GR00T is definitively the superior method. Performance can be influenced by many factors, including the amount of compute used (e.g., batch size), hyperparameters such as learning rate, and whether the visual or language backbones are fine-tuned.
> One possible explanation for the observed gap is that we use a larger batch size for GR00T (128) than for pi0 (64), made possible by GR00T’s smaller memory footprint (GR00T, by default, does not fine-tune the visual encoder, whereas pi0 does). We adopt the default hyperparameter settings for both GR00T and pi0 and use the largest batch size that fits on a single GH200 GPU. It is entirely plausible that with different hyperparameters or greater compute resources, pi0 could match or exceed GR00T’s performance.
>
> > seeing if LoRA finetuning affects performance, both in 4.2 and 4.3, etc.
>
> For the multi-task learning experiments in section 4.2, we ran GR00T with LoRA fine-tuning, trained for the same number of steps, batch size, etc, as the full finetuning variant. The results are as follows:
>
> | Method            | Atomic-Seen | Composite-Seen | Composite-Unseen | Average |
> |-------------------|-------------|----------------|-------------------|---------|
> | **Full finetuning** | 43.0%      | 9.6%          | 4.4%             | 20.0%   |
> | **LoRA finetuning** | 2.4%       | 0.2%          | 0.8%             | 1.2%    |
>
> Full finetuning is critical to model performance. We have added these results to Appendix G.4. We would also like to run LoRA fine-tuning for the foundation model training experiments, however due to compute resource constraints we leave this to the community to try with our eventual open source code release.
>
> > Another suggestion is to stratify the tasks, like contact rich tasks, mobile manipulation tasks, etc. and see performance profiles across different tasks.
>
> Thank you for the feedback. We have added Appendix Section H.1, in which we perform an analysis of the best performing model from the foundation model training experiments. We break down the model’s performance per task, and outline information about the tasks such as the number of stages involved and whether the task involves mobile manipulation. The most clear takeaway is that composite tasks are more difficult to learn than atomic tasks, possibly due to their longer horizon nature and their incorporation of multiple skills. Digging deeper, we do a qualitative analysis of the tasks for which the model performs worst at. These are a set of tasks, some with very long horizons but others with shorter horizons, some involving more simple skills such as pick and place and others involving more dexterous skills such as turning on the stove burner. Overall, there is a long list of failure modes exposing the limitations of the model, such as limited skill reliability, lack of long-term memory, and incorrect semantic reasoning.

---

### Official Review · Reviewer_rSty · 2025-11-01

**Soundness:** 2
**Presentation:** 2
**Contribution:** 2
**Rating:** 6
**Confidence:** 4

**Summary:**

This work introduces RoboCasa365, a simulation benchmark for household robotic manipulation tasks. The benchmark is an extension of RoboCasa, and introduces 365 everyday tasks across 2500 diverse kitchen environments. The authors use this benchmark to evaluate multi-task learning, foundation policy model training, and lifelong learning. Extensive experiment results on the proposed benchmark provide insights into how task diversity, dataset scale, and environment variation influence generalization in robot manipulation.

**Strengths:**

1. The additional assets and scenes significantly increases the task diversity of the original RoboCasa benchmark. The simulated kitchen environments have highly realistic and diverse layouts; and the authors ensure the task activities are from a diverse set of categories.

2. The paper presents systematic experiments to study key factors in training generalist robot policies. The authors have dedicated effort into implementing various policy training baselines and compared results on both seen and unseen tasks.

3. Good presentation quality and easy-to-follow writing. Especially sections 3 and 4 are well-organized and provide a good amount of high-level and low-level details about both the dataset statistics and experiment settings.

**Weaknesses:**

1. Lack of real world evaluations. Although a diverse simulation task suite provides many opportunities for studying algorithms, without a real world digital twin or policy transfer evaluation, the value of the benchmark is limited and it's unclear if any assumptions or implementation bugs in simulation environments would hinder transfer to the real world.

2. Lack of qualitative results. The submission did not include any supplementary materials. For policy learning, it would have been much clearer to show videos of the successful policy rollouts and failure modes. For the task scene designs, it would also improve clarity if the authors can provide more renderings from more camera views into the scenes.

3. In Table 4, the authors claimed "Comparing the Human50 and Human300 settings, we see that increasing the number of pretraining tasks enables a significant improvement in downstream posttraining" -- but the numbers between 2nd and 3rd columns are not really "significantly" different, e.g. 41.0 vs 41.2, or 26.2 vs 28.7. If increasing the number of pretraining tasks from 50 to 300, which is a big change, still yields very minimal performance gains, it makes it questionable whether the training and testing tasks are setup to have enough diversity and/or correlation.

**Questions:**

1. How are the robot arm controllers implemented, and what's the control frequency? What's the action space -- is it end effector poses and gripper open/close?

2. How is the task conditioning implemented in multi-task policy training? The paper mentioned it's language-conditioned, but is there more investigations on how to embed the language inputs or alternative ways to do task conditioning.

---

> ### Author Response · Authors · 2025-11-24
> **Author Response to Reviewer rSty (Part 1/2)**
>
> Thank you very much for your detailed and thoughtful review! We appreciate your positive comments and constructive feedback. We address your comments below:
>
> > Lack of real world evaluations.
>
> Thank you for the feedback. Following your suggestion, we conducted a set of real-world experiments. We assembled a full robotic stack using the DROID Panda arm in a real kitchen environment and collected demonstrations for four tasks:
> - CloseElectricKettleLid: Close the lid of an electric kettle.
> - PickPlaceToasterOvenToCounter: Move an item from the toaster oven to the counter.
> - PickPlaceCounterToCabinet: Move an object from the counter into a cabinet.
> - PlaceOnDishRack: A longer-horizon task involving placing two items from the sink onto a dish rack.
>
> We collected 30 demonstrations for each of the first three tasks and 50 demonstrations for the final task, yielding 140 real-world demonstrations in total. We then compared two training settings:
> - Real-Only: Training GR00T exclusively on the 140 real-world demonstrations.
> - Sim-and-Real (ours): Pre-training GR00T on our simulation tasks (using data from the top half of best-performing simulation tasks), followed by co-fine-tuning on the real-world demonstrations along with analogous simulated data for the four real-world tasks.
>
> After training, we evaluated each model in the real world, running 20 trials per task and reporting success rates. The Real-Only model achieved a mean success rate of 61.8%, whereas our Sim-and-Real approach achieved 79.8%, a substantial improvement. These results highlight the value of our simulation benchmark, not only as a platform for studying and comparing algorithms in simulation, but also as an effective tool for improving real-world policy learning.
>
> | Task                         | CloseElectricKettleLid | PickPlaceToasterOvenToCounter | PickPlaceCounterToCabinet | PlaceOnDishRack | Average |
> |------------------------------|-------------------------|-------------------------------|----------------------------|------------------|---------|
> | **Real-Only**                | 70%                    | 70%                           | 52%                        | 55%              | 61.8%   |
> | **Sim-and-Real (ours)**      | 70%                    | 100%                          | 84%                        | 65%              | 79.8%   |
>
>
> > Lack of qualitative results.
>
> Thank you for the helpful suggestion. We have added supplementary materials, which include renderings of both the pre-training and post-training scenes, videos of the human demonstrations and the MimicGen trajectories, as well as videos of policy rollouts (including both successful and unsuccessful attempts). We provide additional content in the updated Appendix. We show the three camera views that are specifically fed into the policy in Figure 10 in the appendix. We also provide a deeper analysis in Appendix H.1, in which we break down results per task and provide a qualitative assessment of failure modes for tasks for which the model was the least performant.
>
>
> > In Table 4, the authors claimed "Comparing the Human50 and Human300 settings, we see that increasing the number of pretraining tasks enables a significant improvement in downstream posttraining" -- but the numbers between 2nd and 3rd columns are not really "significantly" different
>
> While the performance gap between the Human50 and Human300 settings is less pronounced on seen tasks, we would like to emphasize the differences that emerge on tasks unseen during pre-training—specifically, the Composite-Unseen tasks. When using 10% of the post-training data, Human50 achieves a success rate of 23.8%, whereas Human300 reaches 32.3%, an 8.5% absolute improvement. Even when using the full post-training dataset, Human50 and Human300 obtain 38.5% and 44.0%, respectively—a 5.5% difference that remains meaningful.
>
> These observations suggest that increasing pre-training task diversity provides notable benefits for generalization to unseen tasks, particularly in low post-training data regimes. We have made sure to emphasize these points in the manuscript.
>
>
> > How are the robot arm controllers implemented, and what's the control frequency? What's the action space -- is it end effector poses and gripper open/close?
>
> We adopt the underlying controller from RoboCasa (Nasiriany et al., 2024). Specifically, we use an Operational Space Controller running at 20 Hz that commands the arm through seven action dimensions: three for translation, three for rotation, and one for gripper opening and closing. In addition, we include five action dimensions for mobile base translation and rotation, torso height control, and an action mode that gates mobile base control. We have added this information to Appendix I.1.

---

> > ### Author Response · Authors · 2025-11-24
> > **Author Response to Reviewer rSty (Part 2/2)**
> >
> > > How is the task conditioning implemented in multi-task policy training? The paper mentioned it's language-conditioned, but is there more investigations on how to embed the language inputs or alternative ways to do task conditioning.
> >
> > For all methods, we use language-based task conditioning. Pi0 and GR00T are Vision-Language-Action models and take natural language descriptions directly as task inputs. For Diffusion Policy, we integrate CLIP-derived language features into the ResNet encoder using FiLM conditioning. Additional details are provided in Appendix F.1. At the moment, we are adding support for additional forms of conditioning (eg. conditioning on the name of the task rather than natural language). We are planning to add these to the paper for the final release.

---

> ### Comment · Reviewer_rSty · 2025-11-27
>
> Thank you for the detailed response. I appreciate the effort into providing additional real world results given the limited time during rebuttal period. I don't have further questions or comments for now.

---

### Author Response · Authors · 2025-11-24
**Summary of Rebuttal Revisions**

We are deeply grateful to all the reviewers for their careful consideration of our work and their insightful, constructive feedback. We have updated the manuscript (major changes highlighted in blue) and added supplementary materials. We summarize the major changes as follows:
- Newly added supplementary materials, showing human demonstrations, MimicGen demonstrations, policy rollouts, and scene renderings.
- Real-world experiments (Section 4.5)
- More in-depth analysis of the foundation training model, with qualitative analysis highlighting failure modes on most challenging tasks (Appendix H.1)
- More in-depth discussion of policy learning architectures and training protocols (Appendix F.1)
- Jointly training on pretraining and post-training data in one phase (Appendix G.3)
- LoRA finetuning (Appendix G.4)

We welcome any further questions, concerns, or feedback you may have. Thank you!

---

### Meta-Review · Area_Chair_ciLk · 2025-12-16

**Summary:**

RoboCasa365 extends RoboCasa, providing 365 daily tasks and 2500 kitchen scenarios for evaluating multi-tasking, base models, and lifelong learning. System experiments reveal the impact of task diversity, data scale, and environmental changes on generalization.

**Reviewer Concerns:**

The authors have added demonstrations and real-world experiments based on feedback, supplemented failure case analyses, standardized the two-stage training, added LoRA fine-tuning, and attached visualization materials. All reviewers' comments have been addressed.

**Reviewer Scores:**

The reviewers consistently maintained positive scores.

---

### Decision · Program_Chairs · 2026-01-26

Accept (Poster)